# Comparison between Two Time Synchronization and Data Alignment Methods for Multi-Channel Wearable Biosensor Systems Using BLE Protocol

**DOI:** 10.3390/s23052465

**Published:** 2023-02-23

**Authors:** He Wang, Jianan Li, Benjamin E. McDonald, Todd R. Farrell, Xinming Huang, Edward A. Clancy

**Affiliations:** 1Worcester Polytechnic Institute, Worcester, MA 01609, USA; 2Liberating Technologies, Inc. (LTI), Holliston, MA 01746, USA

**Keywords:** biosensor, BLE (Bluetooth Low Energy), biomedical signal, time synchronization, wireless sensor network

## Abstract

Wireless wearable sensor systems for biomedical signal acquisition have developed rapidly in recent years. Multiple sensors are often deployed for monitoring common bioelectric signals, such as EEG (electroencephalogram), ECG (electrocardiogram), and EMG (electromyogram). Compared with ZigBee and low-power Wi-Fi, Bluetooth Low Energy (BLE) can be a more suitable wireless protocol for such systems. However, current time synchronization methods for BLE multi-channel systems, via either BLE beacon transmissions or additional hardware, cannot satisfy the requirements of high throughput with low latency, transferability between commercial devices, and low energy consumption. We developed a time synchronization and simple data alignment (SDA) algorithm, which was implemented in the BLE application layer without the need for additional hardware. We further developed a linear interpolation data alignment (LIDA) algorithm to improve upon SDA. We tested our algorithms using sinusoidal input signals at different frequencies (10 to 210 Hz in increments of 20 Hz—frequencies spanning much of the relevant range of EEG, ECG, and EMG signals) on Texas Instruments (TI) CC26XX family devices, with two peripheral nodes communicating with one central node. The analysis was performed offline. The lowest average (±standard deviation) absolute time alignment error between the two peripheral nodes achieved by the SDA algorithm was 384.3 ± 386.5 μs, while that of the LIDA algorithm was 189.9 ± 204.7 μs. For all sinusoidal frequencies tested, the performance of LIDA was always statistically better than that of SDA. These average alignment errors were quite low—well below one sample period for commonly acquired bioelectric signals.

## 1. Introduction

Recent studies have shown the rapid development and expanded use of wearable sensor systems for human study, medical research, and healthcare [1,2,3,4,5,6,7,8,9]. Low-power transmission protocols, such as BLE (Bluetooth Low Energy), ZigBee, and low-power Wi-Fi, are commonly applied in various wireless applications, including human rehabilitation, biomedical signal monitoring, and healthcare systems [10,11,12,13]. The monitoring and collection of bioelectric signals such as EEG (electroencephalogram), ECG (electrocardiogram), and EMG (electromyogram) require low power consumption, low latency, high accuracy, and a relatively high bandwidth compared to other biomedical signals [14,15,16]. ZigBee does not meet the bandwidth requirement for these bioelectric signals, especially for EMG signals. Compared with low-power Wi-Fi, BLE has lower power consumption of ~1 mA and a high transmission speed of 2 Mbps [17,18,19]. Due to the rapid upgrade rate of these protocols, we chose standard BLE version 5.0 to develop our wireless wearable biomedical signal system.

Most bioelectric signal applications require multiple channels (and, thus, peripheral nodes) for signal collection and processing [20]. Each peripheral node consists of its own analog-to-digital-converter (ADC) which follows its local clock, and each ADC will run at a slightly different sampling rate and phase [21]. These differences cause asynchronization between the peripheral nodes within the system. However, the native time synchronization methods supported by these protocols cannot satisfy the required high sampling rate of common bioelectric systems [22]. Hence, a well-designed, transferable time synchronization method is needed in order to support various wireless devices.

Previously, several time synchronization methods have been introduced [23,24,25], specifically for use with BLE. In particular, the BLE beacon role or connection events tend to be feasible methods [26,27]. Using the beacon role, the central node periodically broadcasts clock information to the peripheral nodes to maintain high synchronization accuracy (drift of <1 µs/min [26]). However, while broadcasting the clock information, the central node cannot receive data packets from peripheral nodes in real time. Alternatively, time synchronization can occur if multiple nodes are connected near-simultaneously (time synchronization differences of 39.92 ± 14.19 µs [27]). Again, this synchronization method cannot be maintained during data collection, as nodes would need to constantly disconnect and re-connect. Thus, these methods are not feasible for continuous, high-throughput, low-latency applications. Another solution is to implement additional hardware. One such approach provided time synchronization precision of 9 ± 17 µs [28]. However, customized hardware is not transferable for the large variety of commercial devices. The additional hardware also increases power consumption, which decreases the battery life of the wireless wearable system in daily use.

We previously developed a peripheral–central node time synchronization method that operates by exchanging periodically generated timestamp information in the software application layer, without the need for additional hardware. Our method is transferable between software versions and different manufacturers’ devices. We found that a simple data alignment algorithm (SDA) achieved precise time synchronization and data alignment (error of approximately one sample period) in real time [29]. However, this method can only adjust the time difference between peripheral nodes to within one half sample period, at best. In this paper, we describe a linear interpolation data alignment (LIDA) algorithm that can adjust time differences between data samples to precision levels below one half sample.

The primary objective of this study, therefore, was to characterize the data alignment performance of peripheral BLE wireless nodes, contrasting the SDA and LIDA algorithms over a range of input signal frequencies. We hypothesized that the increased temporal resolution of the LIDA algorithm (due to its between-sample interpolation) would produce better data alignment. Of course, high-precision data alignment relies entirely on the performance of time synchronization between peripheral nodes and the central node. Hence, our tests also demonstrate the precision of our time synchronization method. This characterization was completed using two peripheral nodes and one central node as part of a wireless demonstration system operated at the benchtop in our laboratory.

## 2. Materials and Methods

### 2.1. Paired Timestamp Generation and Peripheral ADC Timestamp

A wireless microcontroller (MCU) can query its clock and return a “timestamp”, which is an *integer* count of the number of fixed microsecond-scale clock tick periods that have occurred since device power-up. On many microcontrollers, this timestamp rolls over after several minutes or hours, this roll over duration being based on the clock tick period and the number of bits used to store a timestamp. To avoid rollover, we cast this timestamp into an unsigned 64-bit integer [29]. These timestamps were used to synchronize each peripheral node to the central node’s clock. The challenge of programming this functionality in the application layer—which facilitates transferability between BLE MCUs from various manufacturers—is in finding a manner by which a timestamp from the central node can be “paired” with a timestamp from a peripheral node. Pairing refers to the two timestamps corresponding to either the same actual time, or to times with a consistent time offset between them. So long as all peripheral nodes in a system receive paired timestamps with the same time offset, the various peripheral nodes can be mutually time synchronized.

Generating paired timestamps in BLE systems that are continuously sampling ADC data at rates up to 2000 Hz (e.g., for EMG acquisition) is challenging. Many systems use beacon transmissions to achieve paired timestamps [26,27,30]. However, when the central node broadcasts a beacon transmission, it cannot receive data from peripheral nodes, thus risking interruption of the high-bandwidth, continuous peripheral data streams needed for these bioelectric signals. Data loss could occur. Alternatively, researchers have used custom add-on hardware to detect a synchronizing signal [28]. Custom design is required and is specific to that implemented system. Hence, the system may require a complete re-design as each new hardware microcontroller or BLE software version is released (which occurs quite frequently, i.e., yearly, or even more frequently in some cases).

Thus, we utilized a time synchronization method that produces paired timestamps using methods from within the BLE application layer [29]. When a timestamp pair is desired, this method initiates peripheral–central data exchange from the central node. We achieved this via a central-to-peripheral node BLE notification that can include a small amount of data transmission, sufficient in size to include a central timestamp.

For optimal synchronization of peripheral devices, upon arrival on the central node of every *K*^th^ new data packet from a peripheral node, the central node immediately queries its clock for a timestamp. This timestamp is incremented by one transmit interval and forms an excellent estimate of the central timestamp ($TS_{c}$) corresponding to the *ensuing* BLE notification when *received* on that specific peripheral node. Timestamp $TS_{c}$ is transmitted to that peripheral as part of the ensuing central-to-peripheral notification. Once a notification is received by a peripheral node, it immediately queries its clock for a timestamp, denoted $TS_{p}$. The timestamp pair $TS_{c}$ and $TS_{p}$, to the best of their ability, represent the same times on each of the central and peripheral devices. Because our method generates timestamps every *K*^th^ data packet, the timestamps are generated at a nominally periodic rate.

In addition to this timestamp pair, each peripheral node queries its timestamp clock immediately after an ADC packet has been sampled. This timestamp ($TS_{ADC}$) closely corresponds to the peripheral time associated with the last ADC sample in the packet and is used to align data after time synchronization. The three timestamps can be used on the peripheral node for time synchronization and data alignment and/or returned to the central node (in the ensuing peripheral-to-central data packet) for downstream synchronization and alignment. In this manner, synchronization and alignment can be performed in real time on either the peripheral or central node. In addition, the ADC data and timestamps could be transmitted over the serial port of the central node to a PC for the offline analysis reported herein.

### 2.2. Affine Model Time Synchronization

The *N* most recent timestamp pairs from each peripheral node were used to synchronize their respective times to that of the central clock via an affine regression model that estimated central time from peripheral time. Let $x_{m}\left[n\right],\;0\leq n<N$ be the most recent *N* peripheral timestamps and $y_{m}\left[n\right],\;0\leq n<N$ be the most recent *N* paired central timestamps, where 1≤m≤M indexes the peripheral nodes. The affine model that estimates the *floating point* formatted central timestamp (${\hat{TS}}_{c,m}$) based on peripheral timestamps ($TS_{p,m}$) is:(1)$${\hat{TS}}_{c,m}=\beta_{0,m}+\beta_{1,m}\cdot TS_{p,m}+\varepsilon_{m},$$
where $\beta_{0,m}$ is the intercept parameter for peripheral node m, $\beta_{1,m}$ is the slope parameter for peripheral node m, and $\varepsilon_{m}$ is a random error term. Parameters $\beta_{0,m}$ and $\beta_{1,m}$ were estimated for each peripheral using linear least squares [29,31] as follows:(2)$$\left\{\begin{array}{cc} \beta_{1,m}= & \frac{N\cdot \Sigma_{xy,m}-\Sigma_{x,m}\cdot \Sigma_{y,m}}{N\cdot \Sigma_{xx,m}-\Sigma_{x,m}\cdot \Sigma_{x,m}}\\ \beta_{0,m}= & \frac{\Sigma_{y,m}-\beta_{1}\cdot \Sigma_{x,m}}{N} \end{array},\right.$$
where $\Sigma_{xy,m}=\overset{N-1}{\underset{n=0}{\sum }}x_{m}\left[n\right]\cdot y_{m}\left[n\right]$, etc. We chose the least squares method of parameter optimization because it minimizes the mean squared estimation error, is a non-iterative optimization technique with no expected convergence issues (and is thus robust), and has a low computational cost. Note that slope parameters $\beta_{1,m}$ will have values very close to 1.0 (depending on the accuracies of the central and peripheral node clocks). The affine regression model is updated for each respective peripheral node when each new timestamp pair is received, hence, every *K* transmit intervals.

### 2.3. Simple Data Alignment (SDA) Algorithm

Given a clock model (updated every *K* transmit intervals) that relates time on the peripheral node to time on the central node, and the ADC timestamp for each data packet from a peripheral ($TS_{ADC,m}$), peripheral data samples can now be time-aligned. Our methods align each peripheral device to central time, thus mutually time synchronizing them indirectly to each other. We begin by applying the respective affine clock model to a peripheral data packet’s ADC timestamp, producing an estimate of the central time corresponding to the last sample in the packet (${\hat{TS}}_{ADC,m}$).

Since different peripheral nodes complete the sampling of ADC packets asynchronously, we begin our SDA algorithm by utilizing the estimated central timestamps (${\hat{TS}}_{ADC,m}$) of the very first packets (generated after device power-up and the initialization of time synchronization) from all *M* peripheral nodes to align each peripheral node to the last peripheral node that transmits packets. That is, the estimate of central time produced from the last connected peripheral node serves as the “primary” clock. ADC data samples collected before the arrival of packets from this last recognized peripheral are discarded. This process is shown in Figure 1.

Once initialized, our SDA algorithm need only account for diverging clock rates (and clock variations) between the various peripheral nodes. When we detect that the accumulated time drift between the clock on peripheral m and the clock on peripheral M (the primary clock) has exceeded a time threshold (in our case, equal to one ADC period), we remove or interpolate (as needed) one sample within a packet from peripheral m. In doing so, we entrain time in the data stream from peripheral m to time in the data stream from peripheral M. Since the ADC timestamps from all nodes estimate time on the central node, the data streams from all peripheral nodes closely reflect time on the central node.

In practice, implementing SDA from the least squares estimated ADC timestamps (${\hat{TS}}_{ADC,m}$) is computationally expensive and potentially subject to round-off error, since these estimates of central time are floating point values. However, once data stream alignment is initialized (see Figure 1), we need only track *changes* in the passage of time between peripheral node M and each other peripheral node. We can do so in a more computationally efficient manner using the original timestamp pairs, which are integer values. To do so for peripheral m, we utilize two of its successive timestamps from the central node, $TS_{c,m}\left[i\right]$ and $TS_{c,m}\left[i+1\right]$, and two corresponding paired timestamps from peripheral node m, $TS_{p,m}\left[i\right]$ and $TS_{p,m}\left[i+1\right]$ (i.e., two successive timestamp pairs). We define the central timestamp difference for timestamp i as:(3)$$∆TS_{c,m}\left[i\right]=TS_{c,m}\left[i+1\right]-TS_{c,m}\left[i\right],$$

Similarly, the corresponding peripheral timestamp difference is:(4)$$∆TS_{p,m}\left[i\right]=TS_{p,m}\left[i+1\right]-TS_{p,m}\left[i\right],$$

Ideally, $∆TS_{c,m}\left[i\right]$ = $∆TS_{p,m}\left[i\right],\;\forall \;i$. However, the clock rate of each node within the BLE system will vary, which results in a difference between $∆TS_{c,m}$ and $∆TS_{p,m}$:(5)$$∆TS_{diff,m}\left[i\right]=∆TS_{c,m}\left[i\right]-∆TS_{p,m}\left[i\right],$$

Once the absolute sum of successive $∆TS_{diff,m}\left[i\right]$ values is greater than the threshold (*tc*):(6)$$\left|\underset{i}{\sum }∆TS_{diff,m}\left[i\right]\right|>tc,$$
there are one or more extra or missing data samples. To be specific, if $\sum ∆TS_{diff,m}\left[i\right]>tc$, there are one or more data samples missing; thus, we inserted one data sample into the current packet. If $\sum ∆TS_{diff,m}\left[i\right]<-tc$, there are one or more extra data samples; thus, we removed one data sample from the current packet. We chose to insert/remove one sample per packet for stability. In preliminary testing (partial results available in [32]), we evaluated the threshold values (tc) of 1, 1.1, 1.25, 1.5, 1.75, 2, and 2.5 samples. Thresholds that are too small risk alternately interpolating and deleting values in the data stream (overly sensitive), whereas thresholds that are too large risk retaining true synchronization errors for a longer duration. We found minimal data alignment performance differences as a function of the threshold values tested. Thus, we used a threshold value of tc=1 sample.

This algorithm cannot target the exact moment that the drift occurs; in fact the drift occurs over a range of sample times. Thus, when interpolating an extra data sample, we chose to do so prior to the oldest sample of the current data packet. Similarly, when removing one data sample, the oldest sample of the current data packet was removed.

### 2.4. Linear Interpolation Data Alignment (LIDA) Algorithm

By its design, the SDA algorithm—assuming perfect timestamp pairs—can align the peripheral nodes to a resolution of approximately one half of a sample period, since data alignment corrections consist only of deleting one sample or inserting one sample. In practice, a higher worst-case alignment error may occur, depending on the value of the correction threshold (set to one sample period above). A threshold greater than one sample is used to prevent deletions from being quickly followed by interpolations in the data stream. Such alignment corrections are attributed to being overly sensitive to small timing variations. Conversely, larger thresholds permit misalignment to persist for longer durations.

In some applications (e.g., measures of cross-channel mutual information, such as cross-correlation), finer alignment (e.g., lower than half of a sample period) is desired. Since each data packet has a central timestamp estimate (based on the affine time synchronization model) corresponding to its last ADC sample, this clock can be combined with the known ADC sampling period to assign an estimated central clock time to each ADC sample—not just the last sample in a packet. Over the time span of one packet (herein, 15 ms), this assignment is quite precise. Thus, the interpolation between adjacent sample values in each peripheral stream can be used to resample the original signal periodically at times referenced to a common central clock. Herein, we develop a straight-line approximation of peripheral data sample values at times between the available sample times.

Consider two consecutive data samples, $D_{m}\left[i\right]$ and $D_{m}\left[i+1\right]$ (units of volts or ADC counts), from peripheral node m and their corresponding linear regression-generated timestamps ${\hat{TS}}_{c,m}\left[i\right]$ and ${\hat{TS}}_{c,m}\left[i+1\right]$. These timestamps refer to time on the central node. A straight-line model of the data values between successive samples would require that:(7)Dmi=αmi·TS^c,mi+δmi Dmi+1=αmi·TS^c,mi+1+δmi,
where $\alpha_{m}\left[i\right]$ is the slope and $\delta_{m}\left[i\right]$ is the offset for the interval between the given data samples. From these two equations, we can solve for $\alpha_{m}\left[i\right]$ and $\delta_{m}\left[i\right]$ as follows:(8)$$\alpha_{m}\left[i\right]=\frac{D_{m}\left[i+1\right]-D_{m}\left[i\right]}{{\hat{TS}}_{c,m}\left[i+1\right]-{\hat{TS}}_{c,m}\left[i\right]},$$
(9)$$\delta_{m}\left[i\right]=\frac{D_{m}\left[i+1\right]\cdot {\hat{TS}}_{c,m}\left[i\right]-D_{m}\left[i\right]\cdot {\hat{TS}}_{c,m}\left[i+1\right]}{{\hat{TS}}_{c,m}\left[i\right]-{\hat{TS}}_{c,m}\left[i+1\right]},$$

Given any desired timestamp value within this interval, ${\hat{TS}}_{c,m}\left[j\right]$, the interpolated value of the corresponding data sample, $D_{m}\left[j\right]$, can be calculated as follows:(10)$$D_{m}\left[j\right]=\alpha_{m}\left[i\right]\cdot {\hat{TS}}_{c,m}\left[j\right]+\delta_{m}\left[i\right],$$

Hence, by applying this algorithm, the peripheral data samples can be resampled to time locations selected by the central node, e.g., sequential periodic times in phase with the true central node clock. When the selected central time advances outside of the timestamp range given by ${\hat{TS}}_{c,m}\left[i+1\right]$, the algorithm proceeds to the next timestamp interval (and its corresponding data samples). By aligning the data stream from each peripheral node to the true central clock, the peripheral data streams become mutually aligned. A fundamental advantage of this method is that it does not rely on the availability of a primary clock from one of the peripheral nodes.

### 2.5. Methods of Analysis

The time synchronization and data alignment methods were evaluated using BLE as implemented on the TI CC26XX family of microcontroller (MCU) modules. We used a TI CC2652R1 as the central node and two TI CC2640 modules as the peripheral nodes. Each peripheral incorporated a 12-bit, unipolar (0–3.3 V) ADC, with the sampling rate set to 1000 Hz. Embedded software for the TI CC2652R1 central nodes was developed using the TI SimpleLink software development kit: version 3.10.01.11, compiler version TI v18.12.2 LTS. For the TI CC2640 peripheral nodes, we similarly used a TI software development kit: version 1.40.00.45, compiler version TI v16.9.1 LTS. In each case, we programmed in the “C” language using Code Composer Studio (TI, version 9.0.1). Off-line data analysis was completed using MATLAB (The MathWorks, version R2022a) on a Dell OptiPlex 3010 desktop computer.

A signal generator (Hewlett Packard 33120A) generated sine waves (800 mVp-p, with a 1 V DC offset) as the common analog input to the ADC on each peripheral node. This setup is shown in Figure 2. To explore the performance of the two data alignment algorithms, we varied the frequency of the input sine wave signal between each trial from 10 Hz to 210 Hz in increments of 20 Hz, for a total of 11 frequencies. Almost all the power in the EEG signal is found below 40–50 Hz [33]. For ECG, most of the power is below 30–40 Hz and most diagnostic information is located below 100 Hz in adults (150 Hz in infants) [34,35]. For EMG, most of the signal power lies below 400–500 Hz, with a mode frequency of approximately 70 Hz (depending on the electrode configuration) [36,37,38]. Hence, the frequency range of the input sine wave signals used in our performance tests spans much of the relevant range of these three common bioelectric signals.

Each testing trial lasted 12 min. Each input frequency was recorded 5 times. Hence, there were 55 testing trials in total. The data packet transmit interval (i.e., connection interval) between each peripheral node and the central node was 15 ms (hence, 15 samples per packet). This packet transmit interval minimized latency while reliably maintaining two connected peripheral nodes for these TI CC26XX MCUs. The timestamp transmit interval was 990 ms, that is, one new timestamp update every *K* = 66 data packets. We selected this interval to minimize time synchronization error, based on prior evaluation of this system [29]. We used a UART to transfer data (and timestamps) received at the central node to a PC for offline analysis in MATLAB. In this manner, the same recorded data were separately processed and compared for mutual time alignment using the SDA algorithm and the LIDA algorithm. All other parameters of the two peripheral BLE nodes were the same.

Analysis using the PC began with time synchronization, which used a buffer of timestamp pairs of length *N* = 128, corresponding to synchronization over a time duration of 127 × 990 ms = 125.73 s (approximately 2 min) [29]. After data alignment using the PC via one of the two algorithms, the first 2 min of each trial were removed to account for startup transients in time synchronization. The remaining 10 min were separated into contiguous epochs, each containing 100 sine wave cycles. Thus, epoch duration varied with sine wave frequency. The data within each epoch were upsampled 100 times, and then, the first and last 40 data samples (0.4 ms) were discarded, to account for the startup transients in the upsampling filter [39]. Upsampling increased the time resolution from 1 ms to 10 μs. The epoch mean value was subtracted from each value, separately for each of the two peripheral channels. The cross-correlation coefficient function between the two peripheral channels was then computed from each epoch. We limited the computation to lags over a range of −0.75 to +0.75 times the sine wave period. For each epoch, the absolute value, in ms, of the location of the correlation maximum was the estimate of the absolute time alignment error between data streams from the two peripheral nodes. The mean and standard deviation of the absolute time alignment error, the 90th and 95th percentile absolute time alignment errors, and the corresponding correlation coefficient function maximum value were determined across all epochs per input frequency (i.e., across the time alignment error results from all epochs in each of the five trials per sine wave). Since the correlation function of a sine wave is, itself, periodic in the sine wave period, the limited search range prevented noise from erroneously identifying the maximum at periodic multiples of the true maximum, especially when the input frequency was high. We also made histogram plots of the absolute time alignment errors for each sine wave frequency and one histogram concatenating the absolute time alignment errors from all trials. The histogram bin edges were incremented by 0.1 ms, and the histogram plots were normalized so that the heights of the bins represent probabilities.

## 3. Results

Figure 3 shows a portion of 10 Hz sine waves from two peripheral nodes before any alignment and after applying LIDA. It is obvious that after applying LIDA to the raw input signal, the time alignment difference between them becomes barely noticeable visually.

Figure 4 shows the probability density function estimate derived from the absolute time alignment errors using input frequencies of 10 Hz, 110 Hz, and 210 Hz (a–c) and all input frequencies (d). Despite minor differences, the general shapes of the probability density function estimates at each sine wave frequency exhibit a similar trend. From Figure 3, it is obvious that the errors generated using LIDA tended to be closer to 0 ms than the errors generated using SDA. Using LIDA, 25.6% of the absolute errors were below 0.1 ms, 56.2% were below 0.3 ms, and 92.4% were below 1 ms. Meanwhile, using SDA, 12.2% of the absolute errors were below 0.1 ms, 35.5% were below 0.3 ms, and 86.6% were below 1 ms. Among all 38,115 epochs from the 55 testing trials, the absolute time alignment errors generated by LIDA were lower than those generated by SDA 24,756 times (65%), greater than those generated by SDA 12,797 times (34%), and the errors were tied 562 times (1%). The overall average (±standard deviation) absolute time alignment error generated by LIDA using all epochs was 0.38 ± 0.40 ms, while for SDA, it was 0.5442 ± 0.4481 ms. The correlation coefficients generated by LIDA were higher than those generated by SDA 29,397 times (77%), and lower than those generated by SDA 8718 times (23%). The overall average (±standard deviation) correlation coefficient generated by LIDA was 0.9983 ± 0.0083, and for SDA was 0.9945 ± 0.0261. Thus, both data alignment methods produced extremely high correlation coefficients.

Table 1 and Table 2 show summary performance results comparing the SDA algorithm and the LIDA algorithm vs. the input sine wave frequency. The results list the average (±standard deviation) absolute time alignment error between two aligned channels, the 90th and 95th percentile absolute time alignment errors, and the mean (±standard deviation) maximum correlation coefficients. At each input frequency, the average absolute time alignment errors generated using LIDA (all LIDA average absolute time errors were ≤0.58 ms) were always lower than those generated using the SDA algorithm (all SDA average absolute time errors were ≤0.70 ms). The average error did not trend as a function of input frequency, indicating that the implementation of both algorithms is feasible regardless of the frequency of the input signal (over the range of frequencies tested).

Similarly, the average correlation coefficients generated using LIDA were always higher (closer to 1) than those generated using SDA. However, all average correlation coefficients were very close to 1, with the lowest average correlation coefficient being 0.9919 when using SDA at an input frequency of 210 Hz.

These differences were next evaluated statistically. A Kolmogorov–Smirnov test found the absolute time differences and the correlation coefficients each to be non-normally distributed (*p* < $10^{-5}$). Thus, the Wilcoxon signed-rank test was applied between the SDA and LIDA results at each input frequency value, with Bonferroni–Holm correction for multiple comparisons. For each input frequency, the absolute time differences generated using LIDA were always significantly lower than the differences generated using SDA (*p* = $10^{-5}$ for each test), and the correlation coefficients generated using LIDA were always significantly higher than the correlation coefficients generated using SDA (*p* < $10^{-5}$ for each test).

## 4. Discussion

### 4.1. Overall Performance of SDA and LIDA Algorithms

In this paper, we introduced the LIDA algorithm as a method to achieve higher-accuracy data alignment than the SDA algorithm in a multi-channel wireless wearable sensor system. Our wireless evaluation system consisted of one TI CC2652R1 board as the central node and two TI CC2640 boards, with an ADC sampling rate of 1k Hz each, as the peripheral nodes. For our testing, we varied the frequency of our input sine wave from 10 Hz to 210 Hz. Offline, we compared the performance of LIDA and SDA. After time synchronization and data alignment, the absolute time alignment error and correlation coefficient between the two peripheral nodes were calculated to evaluate the performance of the two algorithms.

For each tested input frequency, the performance of LIDA was always statistically significantly better than that of SDA. In particular, LIDA achieved its lowest average absolute time alignment error of 0.19 ± 0.20 ms at an input frequency of 190 Hz. Meanwhile, SDA achieved its lowest average time alignment error of 0.38 ± 0.39 ms at an input frequency of 10 Hz. Across all tested frequencies, the average time alignment error for LIDA was 0.38 ± 0.40 ms, and for SDA was 0.54 ± 0.45 ms. Hence, the average performance of LIDA across all frequencies was better than the best average performance at any frequency of SDA. Since the absolute alignment errors have large standard deviations, which are close in magnitude to the mean values, we also measured the 90th and 95th percentile errors. For each input frequency, the 90th and 95th percentile errors of LIDA were always less than or equal to those of SDA. For both algorithms, the 95th percentile errors were around 1–2 ms (typically around 1.5 ms). Additionally, these 90th and 95th percentile errors did not tend to vary with the input sine wave frequency. As noted above, another clear advantage of the LIDA algorithm is that data from each peripheral node are independently aligned directly to time on the central node. Data alignment from one peripheral node is not a function of any other peripheral node. For our SDA algorithm, we needed to select one primary peripheral clock to which all other peripheral clocks were aligned. Hence, if that primary clock became unavailable (e.g., due to loss of that peripheral node from the network), the SDA algorithm must be restarted (or otherwise modified mid-alignment). A temporary loss of alignment may result. No such issue occurred with the LIDA algorithm. In the future, it is possible that the SDA algorithm could be altered to time-align each peripheral node directly to the estimated central clock times, in much the same fashion as is performed by the LIDA algorithm. Doing so would simplify the overall system complexity, hopefully at little or no cost to performance.

This finding that LIDA out-performed SDA met our expectations, since the SDA algorithm only inserts or removes one data sample at the head of each data packet. Conversely, the LIDA algorithm interpolates sample values between two samples, providing higher precision, at a relatively low computational cost. Overall, considering that the purpose of our wireless wearable system is to acquire high-frequency bioelectric signals, both SDA and LIDA had small time alignment errors. In particular, for EEG and ECG signals, which usually have a sampling rate below 500 Hz, the 95th percentile errors were less than one sample period; for EMG signals, which usually have a sampling rate of 1000–2000 Hz (or higher), the 95th percentile errors were about 1–4 sample periods.

We might improve interpolation performance by upgrading the algorithm by smoothing interpolated values from additional adjacent data samples (e.g., higher-order polynomial or a spline-fit approach). However, it is unclear if the additional smoothing will provide substantive improvement, or if the achieved improvement would be worth the additional computational cost.

### 4.2. Consideration of Previous Time Synchronization Methods

Previously published methods for BLE time synchronization (reviewed above) have demonstrated lower errors than our own, but have some clear disadvantages compared to our methods. The prior method, with the lowest average (±standard deviation) error of 9 ± 17 µs, requires that custom hardware be added to the MCU system [28]. Its time synchronization performance is impressive, but the custom hardware incurs additional non-recurring engineering design effort which might need to be repeated each time the MCU is updated. New MCUs are being offered frequently, often in timeframes of less than one year, by an array of manufacturers, with newer model performance “leapfrogging” that of existing models. Thus, many users might struggle to repeatedly update their custom hardware to take advantage of the improved MCUs. Additionally, the additional hardware adds weight, volume, and battery consumption to the wearable device. Hence, this option is not preferred by many users. Alternatively, high time synchronization precision of 39.92 ± 14.19 µs was achieved when a connection was established simultaneously for multiple nodes [27]. However, this method would either only provide synchronization once at startup (which would drift over time) or require continuously disconnecting and reconnecting devices, interrupting the data stream. Hence, it is not clear that this method is appropriate for high-bandwidth, continuous data acquisition. The use of a beacon role to broadcast clock information [26] has similar issues—the central node cannot receive data packets while broadcasting. Again, since the broadcast would need to be repeated periodically, high-bandwidth, continuous acquisition is not facilitated. Each of the prior methods fills a distinct niche, as does our method. In particular, our method operates in the application layer, making it easily transferable to updated devices or those of other manufacturers. In fact, we have already transferred our SDA algorithm to run on the Nordic nRF5280 microcontroller unit. Finally, our method’s existing performance is already sufficient for many biomedical signal applications.

Note that in addition to these past time synchronization methods developed specifically for BLE, there exist more traditional methods in wired and wireless networks. None were studied herein as they are not considered appropriate for our application (due to their high bandwidth, low latency, and continuous data stream) [40,41]. A network time protocol (NTP) was developed for large-node wired systems. This typically utilizes a hierarchy of many nodes and complex methods for excluding outlier synchronization information. Due to its large volume of synchronization messages, high computation, and poor energy inefficiency, it is not considered appropriate for BLE systems. Traditional time synchronization (TTS) uses two-way messaging between wireless nodes, assuming those messages can be exchanged in a short time duration. However, BLE message exchange times are not tightly controlled in the small MCUs used in wearable devices, making this technique inappropriate. Reference broadcast synchronization (RBS) is a wireless method based on beacon transmissions and least squares linear regression (to smooth out variations from sequential beacon transmissions). As noted above, a beacon role within BLE would require interruption of the data stream from peripheral to central, and thus, it is not appropriate. The wireless methods known as time-sync protocol for sensor networks (TPSN) and flooded time sync protocol (FTSP) do not operate in the application layer (they operate in the medium access control (MAC) layer). Our method reported herein operates in the application layer, so is easier to program and is transferable between devices and software versions.

### 4.3. Connection Loss and Packet Loss

In our tests, we consistently maintained two peripheral nodes connected to one central node without connection or packet loss. In fact, to the extent possible, we eliminated all other active BLE devices from the environment. Doing so was appropriate, since randomly blocked packets would confound our time and data alignment comparisons.

However, regulating the wireless environment in this manner is not possible in applied usage. Connections can be lost due to overutilization of the BLE frequency band (including noise) or simply because the relevant peripheral node has been powered down or has otherwise voluntarily departed the system. If the departed peripheral node had been providing the primary peripheral clock for our SDA algorithm, a new primary peripheral clock must be established. Switching mid-alignment to another peripheral node is possible, but might cause a small transient change in data alignment performance. For our test system consisting of only two peripheral nodes, this issue was moot, as we purposely maintained connection throughout each data collection trial. Further, if one peripheral node were to depart our test system, only one peripheral node would remain; hence, no data alignment between peripheral nodes would be required. Thus, we did not account for this occurrence. Nonetheless, this issue is relevant for systems consisting of more than two peripheral nodes.

For our laboratory tests, it was convenient to wait for all peripheral nodes to connect to the central node, and then, select the *last* connected peripheral node as the primary clock. Our laboratory work always connected all devices (two, in our case) and maintained the connection for each completed trial. Yet, in fielded systems using the SDA algorithm, it might make more sense to assign the *first* connected peripheral node as the primary clock. In this way, data alignment can begin as soon as a second peripheral node connects to the network. However, a new primary clock would still have to be established if this first node later departed the network.

Again, for the purposes of our testing, we chose to maintain a low-noise environment, resulting in no packet loss. However, the applied use of such a system would not prevent packet loss. In this case, our timestamps can be used to quickly and reliably detect packet loss. Successive ADC timestamps from each peripheral node should ideally differ by one connection interval. Even with clock variation between nodes, actual time differences between successive ADC timestamps on one particular peripheral node should easily be within 1% of the connection interval (albeit with a few exceptions; see below). Thus, we can test these time differences as soon as packets arrive on the central node. For example, with our connection interval of 15 ms, a packet has been dropped if the ADC timestamp difference between successive packets from a given peripheral node is 30 ms. Two packets have been dropped if the ADC timestamp difference is 45 ms, etc. Knowledge of this loss of information can be provided to downstream processing for appropriate action.

### 4.4. Parameters of the Affine Time Synchronization Model

Our affine time synchronization model used the most recent *N* = 128 timestamp pairs, which were updated every *K* = 66 data packets. Since data packets were transmitted every 15 ms, the regression used data that spanned approximately the last 2 min to form each model. We had previously evaluated the performance of the SDA algorithm on this same system as a function of the number of timestamp pairs (*N* ranged from 2–128) and the timestamp update interval (which ranged from 150–1500 ms) [29]. We did not find strong dependence on number of timestamp pairs, but an update interval of 750 ms was best. Using this update interval and only *N* = 2 timestamp pairs in the affine model, our previous work found average (±standard deviation) absolute data alignment errors of 69 ± 71 µs, with a 90th percentile error of 180 µs and a 95th percentile error of 190 µs.

In the experimental work in this manuscript, we opted to use *N* = 128 timestamp pairs in our models, with a similar update interval of 990 ms. Both the work in this manuscript and our prior work acquired test data in a low-noise electromagnetic environment in which we had purposely limited traffic in the BLE frequency band. This low-noise environment may have contributed the finding in our prior work of no strong dependence on the number of timestamp pairs, since we eliminated (or greatly attenuated) blocked transmissions. The main advantage of using several timestamp pairs in a regression model is the ability to obtain an average across their variability, and blocked/delayed transmissions contribute strongly to transmission timing variability. However, fielded systems will need to operate in noisier electromagnetic environments. In such cases, the use of a larger number of timestamp pairs in our affine model should reduce time synchronization and data alignment errors. Hence, we chose to use a larger number of timestamp pairs in the present study.

### 4.5. Limitations and Future Work

Perhaps the most fundamental limitation of this paper comes from the generation of the timestamp pairs. Specifically, the central node generates timestamp $TS_{c}$ (queries its clock and adds a transmit interval) when it receives a data packet from a peripheral node. The precision of $TS_{c}$ will be affected if the transmission from the peripheral to the central and/or subsequent notification from the central to the peripheral is not reliable or blocked. (In a blocked transmission, the packet is resent during the next transmission interval.) The use of affine model time synchronization with a large timestamp buffer size smooths some of these possible deviations, but the performance can still be impaired. The more BLE devices the environment contains, the higher the chance of delayed or blocked transmission. Our future work will aim to develop a method to detect blocked transmissions and manually correct $TS_{c}$ to improve time synchronization and data alignment performance.

While it is ideal if timestamp pairs truly reflect time on the central node, it is sufficient for the mutual synchronization of peripheral nodes that central–peripheral timestamp pairs have the same time offset from time on the central node. Accordingly, it is necessary that the *difference* between successive central timestamps tightly matches the *difference* between successive peripheral timestamps. That is, whenever the recording of a central timestamp is delayed (or advanced), no contribution is made to lowering time synchronization so long as the paired peripheral timestamp is equivalently delayed (or advanced). Nominally, each difference will equal the timestamp update rate, which is the connection interval multiplied by *K* (the number of connection intervals between new timestamp pairs). Unfortunately, we did not archive the timestamp pairs in this study, so could not assess this characteristic in our data.

However, our earlier study on this same system that compared synchronization performance as a function of the number of timestamps pairs (*N*) and the timestamp update interval (*K*) did retain the timestamp pairs [29]. For that study, timestamp pairs were available every 1050 ms (this was the closest available interval to our 990 ms interval) during a 12 min trial (thus, there were 685 paired timestamps in total). Figure 5 separately cross-plots these time *differences* between successive central node timestamps on the *x*-axis vs. time *differences* between paired successive peripheral node timestamps on the *y*-axis for each of the two peripheral nodes. Ideally, each timestamp pair should cross-plot onto the line of agreement (*x*-axis value equals *y*-axis value) and be clustered about 1050 ms. In fact, most do so, but a few noticeable outliers exist. As a measure of how well each timestamp difference pair clustered along the line of agreement, we computed the closest distance (*D*) between each x–y location and the line of agreement as [42]: $D=\left|{∆}_{c}-{∆}_{p}\right|/\sqrt{2}$, where ${∆}_{c}$ is the central node timestamp difference and ${∆}_{p}$ is the peripheral node timestamp difference. Histograms of these distances are also shown separately in Figure 5 for both peripheral nodes. Most distances are under 0.5 ms, but large outliers exist. These large outliers may be associated with rare blocked or delayed transmissions.

A few comments should be noted. First, two outliers from peripheral node 2 exceed 100 ms. For the work described herein, in which ~2 min of timestamp data contributed to the affine time synchronization model, such large outliers could degrade time synchronization for up to ~4 min. In the future, analysis of these central–peripheral timestamp differences (e.g., deviation from the line of agreement or deviation from the expected difference between successive timestamps) might allow for the detection of outlier timestamp pairs. Such pairs could then be discarded. Second, the shapes of the distance histograms in Figure 5 are similar to those of the time synchronization histograms in Figure 4, but with smaller standard deviations. However, the timestamp errors from the two peripherals would likely have an additive influence on time synchronization (as each peripheral node timestamp error grows, so does the time synchronization error). Additive error effects lead to larger standard deviations; this is consistent with our results. Moreover, other factors can contribute to time synchronization errors. Third, further study beyond the scope of this work is warranted. For example, it is likely that successive timestamp differences are correlated. For example, a long-duration timestamp difference resulting from a blocked transition might be followed by a short-duration difference (since timestamps are issued nominally in a periodic fashion), and vice versa. It is unclear whether such correlations are relevant to improving time synchronization performance, e.g., by helping to identify timestamp pairs that might be discarded.

Note that the computational cost of the LIDA method is already a limitation in our own implementation on the TI CC2652R1 central node. For our high-throughput real-time application, it is likely that the central node does not have the capability to complete time synchronization and the LIDA algorithm with more than two peripheral nodes. However, SDA has already been implemented on the central node of our real time system and works reliably. A possible solution for this limitation is to move the LIDA algorithm and data alignment process from the central node to each peripheral node. In this way, the central node will directly receive aligned data from each peripheral node, and the repeated computation on one (central) board will be reallocated to multiple (peripheral) boards. This option, however, would likely drain battery power faster in the peripheral nodes. Another alternative is to move the LIDA algorithm computation away from a floating point implementation. Herein, our online implementation utilized floating point arithmetic. Perhaps fixed point computation would provide sufficient resolution. In addition, since the data values are inherently integer-based, portions of the computation might require fewer computational resources if completed directly using integer arithmetic.

Another limitation of this research is that we only evaluated time synchronization performance using two peripheral nodes. Many bioelectric systems (especially EEG and high-density surface EMG) can include more than 100 recording channels. This type of multi-node evaluation was outside the scope of the present study as our own research is currently focused on EMG systems utilizing two wireless nodes. Of course, the more wireless nodes operate in a system, the higher the likelihood of delayed or blocked transmissions—which can lead to degradation in the quality of the paired timestamps. However, signal transmission latency is also an issue for systems with a larger number of nodes. In the research reported herein, we set our connection interval at 15 ms, which was as short a connection interval as possible and facilitated reliable transmission using the TI CC26XX MCUs. As more peripheral nodes are added, the connection interval will need to grow (at least linearly, perhaps at a steeper pace due to the increased packet size), since the central node must receive data from each peripheral node in a serial fashion (i.e., time-division multiplexed). Longer latencies occur, which can be detrimental to real-time control applications such as myoelectric prosthesis control [43] or brain–computer interfaces. In these situations, it might prove advantageous to wire several adjacent bioelectric channels to a single peripheral node (e.g., the TI CC26XX MCUs provide eight ADC channels), thereby reducing the total number of required peripheral nodes. ADC samples from the multiple channels could be bundled into a single packet. Of course, packet size limitations will also constrain such a design. If too many channels are wired to the same peripheral, the number of samples per connection interval may exceed the packet size. In any case, our general approach to time synchronization and data alignment should remain useful. However, the options on how to configure such a system are growing exponentially. Additional research into multichannel systems is warranted.

## 5. Conclusions

In this paper, our primary objective was to contrast the performance of our two data alignment methods (SDA and LIDA), both of which utilized time synchronization operating in the BLE application layer. Both methods were tested using one BLE central node and two peripheral nodes, forming a prototype wireless wearable biosensor system. Since the LIDA method aligns data to a fraction of a sample period, we had hypothesized that it would perform better. The best performance of the SDA method occurred when the input test frequency was 10 Hz, providing an average (±standard deviation) absolute data alignment error of 384.3 ± 386.5 µs (90th percentile error of 0.87 ms). For the LIDA method, the best absolute error performance of 189.9 ± 204.7 µs (90th percentile error of 0.50 ms) occurred when the input test frequency was 190 Hz. Statistical comparisons found that the error using LIDA was lower than the error using SDA for every sine wave frequency tested (*p* = $10^{-5}$ for each test; see also Table 1 and Table 2). Hence, the interpolation provided by the LIDA algorithm was shown to reduce data alignment error compared to the SDA algorithm. Nevertheless, each method provided average (±standard deviation) data alignment errors ≤0.70 ms, which is sufficient for many target applications, including many EEG, ECG, and EMG acquisition systems.

## Figures and Tables

**Figure 1 sensors-23-02465-f001:**
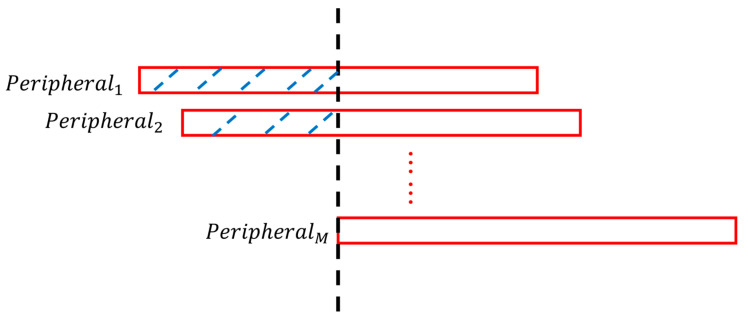
Aligning the M peripheral nodes at startup. Red blocks represent the data packet streams. The black dashed line marks the arrival of the last peripheral into the configuration, signifying the start time of the whole system. Blue dashed lines mark discarded ADC data samples.

**Figure 2 sensors-23-02465-f002:**
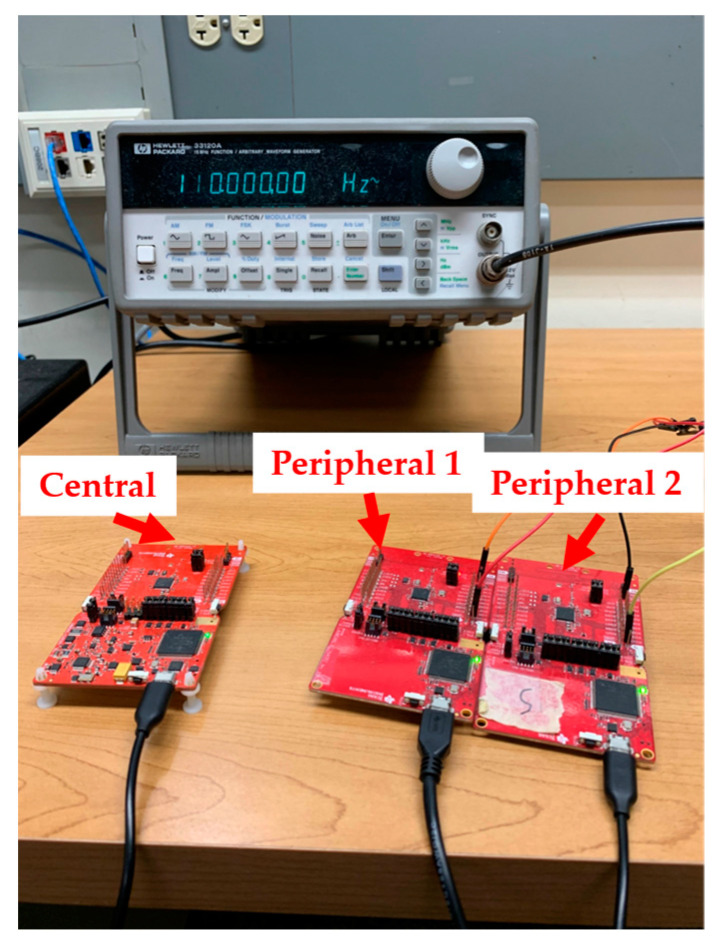
TI platform (one central node and two peripheral nodes) and the Hewlett Packard 33120A signal generator. The signal generator sine wave output is simultaneously connected to the ADC input of both peripheral nodes.

**Figure 3 sensors-23-02465-f003:**
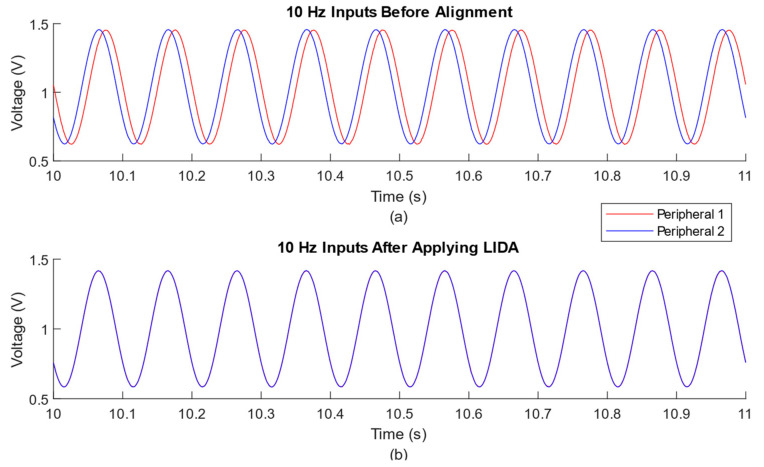
Example time-series plots from one trial (10 s to 11 s time range) of 10 Hz input frequency. (**a**) Input signals from two peripheral nodes without alignment; (**b**) same signals after applying LIDA.

**Figure 4 sensors-23-02465-f004:**
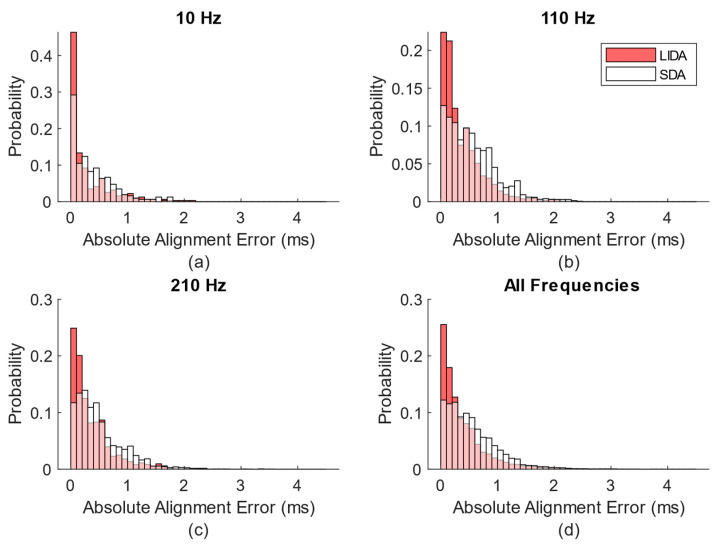
Histogram plot generated from absolute time alignment errors using input frequencies of 10 Hz, 110 Hz, and 210 Hz (**a**–**c**) and all input frequencies (**d**). The red bins were generated using LIDA and the white bins were generated using SDA, with the overlapped parts colored pink.

**Figure 5 sensors-23-02465-f005:**
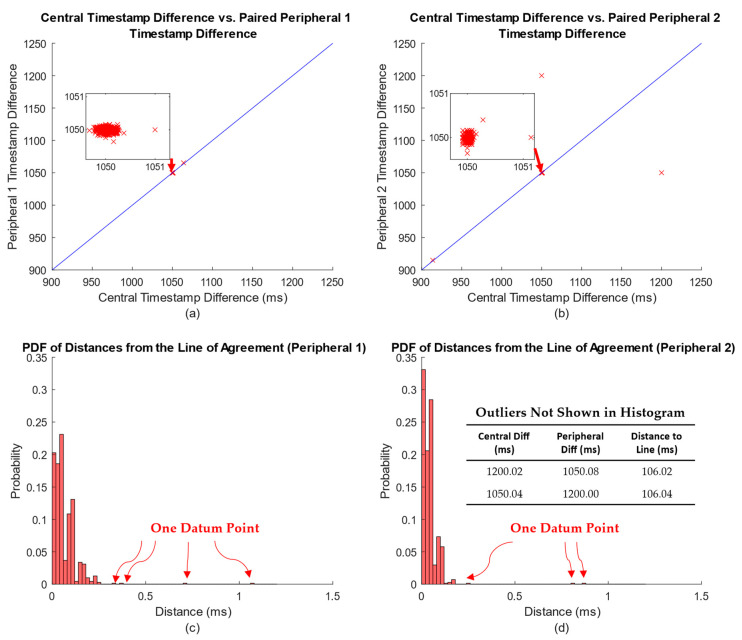
(**a**,**b**) Cross-plots (one per peripheral node, as labeled) of time differences between successive central node timestamps on the *x*-axis vs. time differences between paired successive peripheral node timestamps on the *y*-axis. Inset plots show expanded view of points clustered around the nominal time differences of 1050 ms. Line of agreement drawn in each cross-plot. (**c**,**d**) Corresponding histograms of the distances from each x–y timestamp location and the closest point on the line of agreement. Scales differ in all plots.

**Table 1 sensors-23-02465-t001:** Summary of absolute alignment errors and correlation coefficients of simple data alignment (SDA) algorithm as a function of input sine wave frequency.

Input Frequency (Hz)	Absolute Alignment Error (ms)			Ave. ± Std. Dev. Correlation Coeff.
	Ave. ± Std. Dev.	90th%	95th%	
10	0.38 ± 0.39	0.87	1.19	0.9997 ± 0.0002
30	0.70 ± 0.73	1.53	2.03	0.9991 ± 0.0016
50	0.64 ± 0.59	1.39	2.01	0.9983 ± 0.0031
70	0.59 ± 0.53	1.23	1.61	0.9974 ± 0.0074
90	0.66 ± 0.50	1.40	1.62	0.9966 ± 0.0094
110	0.55 ± 0.43	1.13	1.37	0.9959 ± 0.0125
130	0.49 ± 0.35	0.95	1.10	0.9954 ± 0.0183
150	0.53 ± 0.44	1.17	1.35	0.9947 ± 0.0211
170	0.63 ± 0.48	1.26	1.52	0.9937 ± 0.0260
190	0.45 ± 0.31	0.88	0.98	0.9930 ± 0.0345
210	0.50 ± 0.41	1.06	1.29	0.9919 ± 0.0404

**Table 2 sensors-23-02465-t002:** Summary of absolute alignment errors and correlation coefficients of linear interpolation data alignment (LIDA) algorithm as a function of input sine wave frequency.

Input Frequency (Hz)	Absolute Alignment Error (ms)			Ave. ± Std. Dev. Correlation Coeff.
	Ave. ± Std. Dev.	90th%	95th%	
10	0.30 ± 0.41	0.87	1.15	0.9999 ± 0.00008
30	0.58 ± 0.63	1.29	1.82	0.9997 ± 0.0004
50	0.47 ± 0.57	1.22	1.75	0.9996 ± 0.0008
70	0.41 ± 0.52	1.10	1.48	0.9995 ± 0.0010
90	0.52 ± 0.46	1.21	1.50	0.9991 ± 0.0016
110	0.37 ± 0.37	0.84	1.07	0.9992 ± 0.0020
130	0.34 ± 0.30	0.70	0.87	0.9984 ± 0.0116
150	0.38 ± 0.37	0.85	1.08	0.9988 ± 0.0023
170	0.48 ± 0.42	1.07	1.35	0.9983 ± 0.0043
190	0.19 ± 0.20	0.50	0.63	0.9965 ± 0.0140
210	0.36 ± 0.38	0.83	1.19	0.9978 ± 0.0107

## Data Availability

The data presented in this study are available on request from the corresponding author.

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
