# Peer review of "Comparison between Two Time Synchronization and Data Alignment Methods for Multi-Channel Wearable Biosensor Systems Using BLE Protocol"

_sensors, 2023, doi:10.3390/s23052465_

Round 1

Reviewer 1 Report

The paper presents time synchronization and data alignment methods (SDA and LIDA) for multichannel wearable Biosensor system and the implemented algorithms make use of the BLE application layer. The design and experiment results are encouraging.  

- Authors needs to give more clarity on biosensor (EEG (electroencephalogram), ECG (electrocardiogram) and EMG (electromyogram) usage in this study, it has been mentioned that conducted experiments algorithms using input sinusoidal signals at different frequencies (10 to 210 Hz in increments of 20 Hz—frequencies spanning much of the relevant range of EEG, ECG and EMG signals). Authors should give details of biosensor signal details used in the experiments for each sensor separately.   

- The proposed model uses one central node and two peripheral nodes for experimental analysis. A simulation with multiple BLE devices would be preferable for testing time synchronization and data aliment methods. It would be preferable if the author provided clarification on the system's scalability in the future scope, even though the specifics of the problem are discussed in limitations and future work in terms of delayed or blocked transmission. 

- Authors can improvise the conclusions by including a few more facts about the statistical results.  

- The article can be revised with an additional number of citations. 

Reviewer 2 Report

Dear authors:

The article is interesting, but the objective should be clarified and a section should be added to define the research unit and the research questions and subquestions to be answered in the conclusions.

In the discussion section, the contrast with other similar experiments that ratify or diverge the results obtained is missing.

The instruments and software used should have been made more explicit and "Materials and Methods" should have been added instead of "Methods" in the title of the section.

The conclusion is very poor. It looks like the abstract. It should include the answers to the research sub-questions (if any), paraphrasing the results and their interpretation.

Reviewer 3 Report

First of all, I don't believe the submitted work belongs to the category: biosensors. It falls well outside this category.

Concerning the evaluation of the scientific content (provided by colleague expert in transmission engineering) of the paper, the authors consider the issue of time synchronization for biomedical data acquisition applications, where the existing synchronization algorithms are not able to guarantee high throughput with low latency, portability and low energy consumption. In particular, the authors propose the Simple Data Alignment algorithm for BLE connectivity, which has been additionally enhanced by the Linear Interpolation Data Alignment (LIDA) algorithm. The authors are able to reach very low synchronization errors (lower than 1 ms). From a deep reading of the paper, I can say that it is well structured and all sections give to the reader the possibility to well understand its content.  However, The authors should address the following suggestions before it is consider further for publication.

1)      There is no comparison of the proposed SDA and LIDA algorithms with other existing schemes (e.g. Affine Time Synchronization, TTS, RBS, TPSN, etc.): I suggest the authors to consider some of the existing synchronization approaches, to implement them and compare their performance with the ones of SDA and LIDA;

2)      The readers cannot understand how the threshold in eq. (6) is evaluated. What are the performance of SDA in function of tc?

3)      Is there a particular reason for which the terms in eq. (2) are evaluated through the linear least squares?

Round 2

Reviewer 1 Report

All the suggested comments are well reflected in the revised paper and it can be proceed with further process.  

Reviewer 2 Report

Dear authors:

Thank you ! The corrections have been made and, in my opinion, the article is now more consistent. Best Regards